# Pulmonary Artery Systolic Pressure and Cava Vein Status in Acute Heart Failure with Preserved Ejection Fraction: Clinical and Prognostic Implications

**DOI:** 10.3390/diagnostics13040692

**Published:** 2023-02-12

**Authors:** Gaetano Ruocco, Filippo Pirrotta, Christian Mingiano, Guido Cavati, Cristina Tavera, Alberto Palazzuoli

**Affiliations:** 1Cardiology Unit, “Buon Consiglio” Fatebenefratelli Hospital, 80123 Naples, Italy; 2Internal Medicine Unit, Department of Medical Surgical, Neuro Sciences University of Siena, 53100 Siena, Italy; 3Cardiovascular Diseases Unit, Cardio Thoracic and Vascular Department, Le Scotte hospital University of Siena, 53100 Siena, Italy

**Keywords:** pulmonary artery systolic pressure, inferior cave vein, congestion, HFpEF, acute heart failure

## Abstract

Background: Peak tricuspid regurgitation (TR) velocity and inferior cava vein (ICV) distention are two recognized features of increased pulmonary artery pressure (PASP) and right atrial pressure, respectively. Both parameters are related to pulmonary and systemic congestion and adverse outcomes. However, few data exist about the assessment of PASP and ICV in acute patients affected by heart failure with preserved ejection fraction (HFpEF). Thus, we investigated the relationship existing among clinical and echocardiographic features of congestion, and we analyzed the prognostic impact of PASP and ICV in acute HFpEF patients. Methods and Results: We analyzed clinical congestion PASP and ICV value in consecutive patients admitted in our ward by echocardiographic examination using peak Doppler velocity tricuspid regurgitation and ICV diameter and collapse for the assessment of PASP and ICV dimension, respectively. A total of 173 HFpEF patients were included in the analysis. The median age was 81 and median left ventricular ejection fraction (LVEF) was 55% [50–57]. Mean values of PASP was 45 mmHg [35–55] and mean ICV was 22 [20–24] mm. Patients with adverse events during follow-up showed significantly higher values of PASP (50 [35–55] vs. 40 [35–48] mmHg, (*p* = 0.005) and increased values of ICV (24 [22–25] vs. 22 [20–23] mm, *p* < 0.001). Multivariable analysis showed prognostic power of ICV dilatation (HR 3.22 [1.58–6.55], *p* = 0.001) and clinical congestion score ≥ 2 (HR 2.35 [1.12–4.93], *p* = 0.023), but PASP increase did not reach statistical significance (*p* = 0.874). The combination of PASP > 40 mmHg and ICV > 21 mm was capable of identifying patients with increased events (45% vs. 20%). Conclusions: ICV dilatation provides additional prognostic information with respect to PASP in patients with acute HFpEF. A combined model adding PASP and ICV assessment to clinical evaluation is a useful tool for predicting HF related events.

## 1. Introduction

Right ventricular (RV) dysfunction and pulmonary hypertension are well recognized factors related to the increased risk of adverse event occurrence in patients affected by acute heart failure (AHF) [1,2,3]. However, less is known about RV adaptations in subjects with heart failure with preserved ejection fraction (HFpEF). Indeed, mechanisms of pulmonary congestion and RV adaptation may differ from patients with heart failure with reduced ejection fraction (HFrEF) [4]. Therefore, in acute settings, the congestion occurrence and associated effects on the right side chamber and inferior cava vein (ICV) are less studied. Many reports have documented the importance of detecting specific functional and morphological measurements of RV, but specific analysis and differentiation occurring across different left ventricular ejection fraction (LVEF) values are still lacking [3,5,6]. Nevertheless, a recent position paper indicates the same echocardiographic parameters of RV assessment for both HFpEF and HFrEF [7]. Undoubtedly, echocardiography is an important tool to analyze RV performance and non-invasive hemodynamic conditions during congestion status. Additionally, it provides important insights on pulmonary and systemic congestion. Indeed, the combined assessment of pulmonary artery systolic pressure (PASP) and ICV dimension and collapse may imply a pulmonary and systemic congestion degree beyond RV structure and function [1,8]. In this context, tricuspid regurgitation (TR) may play an additive role: more severe TR degree could subtend a more impaired functional status and increased RV dimension. Accordingly, peak TR velocity demonstrated a good concordance with invasive measurements, and it is now known as a reliable non-invasive method to quantify PASP [9]. Elevated TR velocity and severity are well-recognized features of poor outcomes in the general population and in patients with HFrEF. They are associated with more advanced New York Heart Association (NYHA) class reduced exercise tolerance and recurrent hospitalization [10,11]. Recent studies have demonstrated that ICV dilatation is another index of systemic congestion, and it strictly reflects right atrial pressure [12,13].

However, most of these findings come from HFrEF trials and chronic patients whereas less is known in acute patients affected by HFpEF [4,14]. Notably, we aim to investigate clinical and echocardiographic features in acute HFpEF patients according to TR velocity and ICV dimension. Therefore, we analyzed the prognostic impact of PASP and ICV dilatation across 180 days of follow up after discharge.

## 2. Materials and Methods

### 2.1. Study Population

We evaluated 225 consecutive patients with acute HFpEF admitted to our department from September 2016 to July 2020. Of these 225 patients, 24 were excluded for poor acoustic window, 7 for incomplete echo data and 21 were lost during the follow up. Patients were enrolled within 12 h of hospital admission with diagnosis of new-onset or decompensated acute HFpEF, with signs and symptoms of acute decompensated HF and elevated levels of natriuretic peptides (B-type natriuretic peptide (BNP) > 100 pg/mL). Patients were defined as having HFpEF if LVEF was ≥50% associated with relevant structural heart disease and diastolic dysfunction, according to the recent echocardiographic criteria. Diagnosis of HFpEF was made if 3 of the 5 main criteria were satisfied [15].

We previously excluded patients from the study with reduced LVEF of less than 50%, recent heart valve replacement and coronary artery bypass graft (<3 months), history of pulmonary embolism, idiopathic pulmonary hypertension, neoplastic, hematologic and immune diseases with systemic involvement and a history of pneumothorax and/or lobectomy.

### 2.2. Physical Examination and Blood Tests

Patients were evaluated by two physicians at admission and discharge to assess grade of clinical congestion (congestion score), giving 1 point for each of following signs: pulmonary rales, third heart sound, jugular venous distention, peripheral edema and hepatomegaly (5 total points). A blood test was taken for the measurement of BNP (Biosite Inc., San Diego, CA, USA) at admission and collected in sterile tubes containing EDTA for AHF diagnosis confirmation. Patients with BNP values < 100pg/mL were excluded.

### 2.3. Echocardiography

All echocardiographic measurements were performed by two expert cardiologists (AP and GR) according to the instructions provided by the American Society of Echocardiography [16]. Therefore, principal measurements were recorded and independently reviewed by two distinct physicians. The systolic and diastolic volumes and ejection fraction were determined using apical two and four chamber views by Simpson biplane formula. We evaluated three consecutive cardiac cycles to obtain average pulsed Doppler transmitral flow velocity during early diastole velocity (E wave) and late diastole velocity (A wave) ratio (E/A) and the deceleration time (DT) of E. Placing the cursor laterally and medially at the mitral annulus level, we estimated mitral annulus movement by apical four-chamber tissue Doppler imaging (TDI). We obtained the recordings of systolic peak velocity (S’), early diastolic myocardial velocity (e’) and atrial systole velocity (A’), for three consecutive cardiac cycles. We calculated the ratio of peak early diastolic filling velocity and septal tissue Doppler early diastolic velocity (E/e’). In patients with atrial fibrillation, we measured E/e1 ratio and DT values. E/e’ > 15 was considered an index of elevated left ventricular (LV) filling pressure and severe diastolic dysfunction [17]. The tricuspid annular plane systolic excursion (TAPSE) was obtained by placing the M-mode cursor laterally to the tricuspid annulus. We estimated the PASP by continuous Doppler at tricuspid valve level, and PASP was obtained as the sum of the mean of 4 peak velocity of tricuspid regurgitation and the estimate of right atrial pressure based on ICV diameter and collapsibility. ICV dimension and collapse was calculated by subcostal view measuring in m-mode the internal vessel diameter soon before right atrium (RA) entrance during both expiration and inspiration. When the diameter reduction in inspiration was less than 50%, we added 10 mmHg to the TR velocity [18].

### 2.4. Follow-Up

Patients were followed up for six months after discharge by serial check-up, by telephone or internet interview. Composite outcomes were considered as the sum of death and re-hospitalization cases for heart failure (HF) and mortality due to cardiovascular causes.

### 2.5. Endpoints and Study Goals

In this study, we sought to evaluate the following: 1—the relationship between PASP and ICV and clinical congestion signs; and 2—the prognostic role of increased PASP and ICV diameter in a single and combined model during the 6-month follow-up period.

### 2.6. Statistical Analyses

Continuous variables were expressed as median and inter-quartile range and categorical variables as counts or percentages, and differences in patients with or without adverse event occurrence were tested using the Mann–Whitney non-parametric test and *X*^2^ tests. Congestion score differences between patients with/without adverse event occurrence were analyzed through T-test. Cox univariate and multivariable regression analysis were used to assess the relationship between variables and outcomes. All continued variables included in the Cox regression analysis were transformed into categorical ones according to specific cut-offs. Event-free survival was estimated by the Kaplan–Meier method. All reported probability values were two-tailed, and a *p* value < 0.05 was considered statistically significant. Statistical analysis was performed using the SPSS 20.0 statistical software package (SPSS Inc., Chicago, IL, USA).

## 3. Results

A total of 173 HFpEF patients were included in the analysis. The median age of our population was 81 [77–86] years, and men were 52% of the total sample. Hypertension and atrial fibrillation prevalence were, respectively, 73% and 61%. The median admission BNP was 668 [373–987] and median LVEF was 55 [50–57] %. Mean PASP and mean ICV were 45 [35–55] mmhg and 22 [20–24] mm, respectively. All the other echo and clinical parameters are reported in Table 1.

Compared to patients without adverse event occurrence, patients with 180 days of adverse event occurrence showed significantly higher values of PASP (50 [35–55] vs. 40 [35–48] mmHg, *p* = 0.005) and RV end-diastolic diameter (RVEDD) (43 [38–47] vs. 37 [36–42] mm, *p* < 0.001). Similarly, patients who experienced poor prognosis demonstrated significantly increased values of ICV (24 [22–25] vs. 22 [20–23] mm, *p* < 0.001) and E/e’ (16 [14–18] vs. 14 [12–15], *p* < 0.001). TAPSE was significantly reduced (18 [15–20] vs. 20 [18–22] mm, *p* < 0.001) in patients with adverse event occurrence. Among left ventricle (LV) variables, none reached statistical significance between the groups (Table 1).

Univariate analysis showed that increased PASP (≥40 mmHg) was not significantly related to prognosis (HR 1.60 [0.91–2.82], *p* = 0.106). Conversely, ICV dilatation (>21 mm) and clinical congestion score ≥ 2 were both significantly related to poor outcome (respectively HR 2.75 [1.49–5.07], *p* = 0.001 and HR 4.03 [2.15–7.57], *p* < 0.001). After adjustment for age, gender, cardiovascular risk factors (hypertension, dyslipidemia, diabetes, coronary artery disease, smoke), atrial fibrillation, left ventricular ejection fraction, E/e’ ≥ 15 and admission BNP values, multivariable analysis confirmed prognostic power of ICV dilatation (HR 3.22 [1.58–6.55], *p* = 0.001) and clinical congestion score ≥ 2 (HR 2.35 [1.12–4.93], *p* = 0.023). The PASP increase did not reach statistical significance (*p* = 0.874) [Table 2].

The prevalence of patients with congestion score ≥ 2 or <2 in patients with PASP ≥ 40 mmhg (67% vs. 33%; *p* = 0.001) and in patients with ICV > 21 mm (68% vs. 32%; *p* = 0.001) is shown in Figure 1.

Following the previous findings, we divided our population into four subgroups: 1-ICV > 21 mm with PASP < 40 mmHg, 2-ICV ≤ 21 mm with PASP ≥ 40 mmHg, 3-ICV > 21 mm with PASP ≥ 40 mmHg and 4-ICV ≤ 21 mm with PASP < 40 mmHg. Adverse event rate was significantly higher in groups 1 and 3 with respect to the other groups (50% and 45% vs. 22% and 20%, *p* = 0.013) (Figure 2).

Univariate analysis of four groups confirmed the significant relationship of ICV dilatation (group 1 and 3) with poor outcomes (respectively, HR 3.11 [1.17–8.29], *p* = 0.023 and HR 2.82 [1.32–6.01], *p* = 0.007). Kaplan–Meier survival curves confirmed this trend (Logrank Test *p* = 0.009) (Figure 3).

## 4. Discussion

Clinical assessment evaluating peripheral and central signs of congestion is not accurate enough, and all clinical congestion scores demonstrated modest accuracy [19]. Echocardiography adds important insights by non-invasive estimation of pulmonary and RA pressures that are directly connected with central vein system [20]. Our findings confirmed the additive diagnostic role of both PASP increase and ICV distention evaluated by echocardiography in congestion detection in patients with HFpEF. Therefore, ICV, but not PASP, appears to be related to increased risk, but a combined model including both echo parameters for clinical congestion score is additive for risk stratification. The current data are opposite with respect to other studies, indicating that PASP has significant prognostic relevance, and it is related to more advanced cardiac dysfunction. However, most of these results come from chronic patients in which the permanent PASP elevation may configure a different pattern. Moreover, in HFpEF patients, the PASP increase may have a different impact compared to HFrEF. Additionally, in acute HF, the role of PASP may diverge from the chronic setting. Indeed, the role of decongestion therapy and good diuretic response may significantly reduce PASP value across hospitalization and the prevalence of “dry” discharge could be associated with PASP normalization. High congestion burden and uncomplete resolution during acute management are associated with adverse outcomes [21,22]. However, poor agreement has been demonstrated between congestion status and hemodynamic evaluation [23]. Invasive measurement of pulmonary wedge pressures and central venous pressure are the gold standard to motorize acute patients, but they are currently measured in only a small percentage of patients in intensive care units (ICUs). Moreover, the ESCAPE trial did not demonstrate a significant benefit in invasive monitoring assessment with respect to non-invasive evaluation [24]. Echocardiographic examination represents the most available method, giving several indicators about wedge pulmonary pressure, PASP estimation, RA pressure and hemodynamic status [25]. Notably, the combination achieved between clinical presentation and ultrasound data provides useful information on whole congestion status and primary cardiac dysfunction. Indeed, the simple clinical examination tends to underestimate the effective congestion status, and poor concordance exists between hemodynamic and systemic congestion [26,27]. This is probably due to different mechanisms leading to central and peripheral congestion: the former is closely related to increased LV filling pressure, venous pressure and fluid accumulation into pulmonary circulation district; conversely, systemic congestion is mainly associated with increased sympathetic overdrive, systemic vasoconstriction and capillary vessel dysfunction. Therefore, neurohormonal activation at kidney levels causing Na and water tubular resorption and interstitial perivascular district disarray are two relevant factors [28]. Similarly, a recent document from the Association for Acute Cardiovascular Care (ACVC) reclassifies “wet” patients into three subgroups: pulmonary congestion resulting in acute respiratory failure; systemic congestion related to systemic volume overload; and tissue hypoperfusion leading to multi organ damage [29]. Nevertheless, a study comparing echocardiographic findings of congestion in acute HFpEF vs. HFrEF showed a similar profile in terms of RV dysfunction and venous congestion [30]. Conversely, a large cohort trial measuring pulmonary pressure in all HF categories revealed that elevated PASP is significantly related to readmission in HFpEF patients [31]. According to the current framework, our findings confirm that the coexistence of both situations (elevated PASP and venous pressure) is related to more adverse outcomes in HFpEF. Despite several noninvasive diagnostic methods having been described to assess PH and RA in chronic conditions, few data are reported in acute settings and in HFpEF [4,32]. Notably, a recent HFA position paper introduced a detailed right ventricle (RV) evaluation in HFpEF, although the echocardiographic cutoff reported is based on studies in HFrEF, and it is not supported by cross sectional data [7]. In line with our results, Bosh et al. found that reduced TAPSE/PASP ratio and longitudinal strain were associated with poor outcomes but not increased PASP [33]. Similarly, in a recent study, Dammassa et al. did not find an association between PASP estimated by pulmonary acceleration time and outcome in hospitalized ICU patients [34]. Conversely, a recent study showed that TR and TAPSE/PASP ratio reduction had time-prognostic significance in HFpEF patients that had been recently discharged [35]. Nonetheless, there are contrasting findings about PASP relevance: a recent Vexus score including Doppler analysis of renal epatic and portal districts together with ICV measurement appears strictly related to congestion severity [36]. Additionally, ICV dilation is a marker of high mortality risk in patients admitted for ADHF and associated with impaired renal function [12], although the authors did not report their findings according to LVEF. Of note, a recent single-center study demonstrated a good correlation among ICV, jugular vein diameter distention and N-terminal pro B-type natriuretic peptide (NTproBNP) [37]. However, a recent multicenter study on 505 acute HFpEF patients showed that the discharge assessment of E/e’, TR and ICV collapsibility < 50% was related to poor prognosis independently from atrial fibrillation presence. [38]. Although multiple-site ultrasound evaluation is recommended to better identify multi-organ water retention, few reports have contemporarily examined pulmonary and abdominal districts. Finally, the multiple ultrasound scan assessment is often restricted to chronic patients and to patients with systolic dysfunction [20,39]. Notably, our findings add new insights in acute HF patients, extending the message to patients with preserved LVEF.

## 5. Limitations

Our data are limited due to the retrospective nature of the current analysis. Therefore, the total number of patients enrolled is relatively low, and this is not conducive to achieving definitive results. However, to the best of our knowledge, this is the largest study analyzing the impact of PASP and ICV on outcomes in an acute HFpEF setting. Current measurements should be repeated before discharge, and variation across hospitalization may better recognize patients with residual congestion at higher risk. Moreover, we did not perform cardiac magnetic resonance (CMR), which is one of the better techniques to evaluate RV. However, the high heart rate and the high respiratory rate of HFpEF patients in AHF phases did not allow us to perform this examination. Additionally, during the acute phase, these patients take continuous oxygen supply and infusion therapy. Under these conditions, it is hard to keep patients in a supine position for long time period [40]. The observational follow-up period is restricted to the first 6-month period and a longer follow-up could help to obtain more consistent results. The relationship found between PASP and ICV is not currently reproducible in chronic conditions in which elevated pulmonary pressure and central vein dilatation may have different behavior. A multi-center analysis contemporarily evaluating clinical congestion, PASP, and ICV may lead to a better risk stratification. Therefore, a head-to-head comparison in acute patients according to different LVEF values could be useful for the identification of specific congestion pattern and ultrasound features responsible for adverse outcomes in each AHF subtype.

## 6. Conclusions

Congestion is the most common driver for HF hospitalization, although less is known about congestion appearance in acute patients affected by HFpEF. The combination of PASP elevation and ICV dilatation adds new insights about the fluid retention status in pulmonary and abdominal districts, respectively. ICV dilatation provides additional prognostic information, whereas a PASP increase did not. A model including clinical congestion to PASP and ICV assessment may be achieved for HF-related event prediction in acute HFpEF.

## Figures and Tables

**Figure 1 diagnostics-13-00692-f001:**
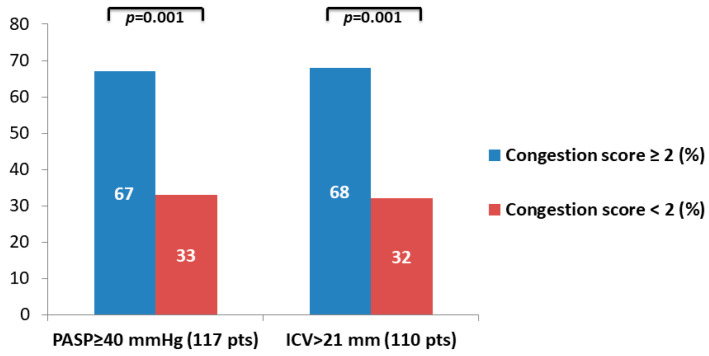
Congestion score ≥ 2 prevalence in patients with PASP ≥ 40 mmhg and in patients with ICV > 21 mm.

**Figure 2 diagnostics-13-00692-f002:**
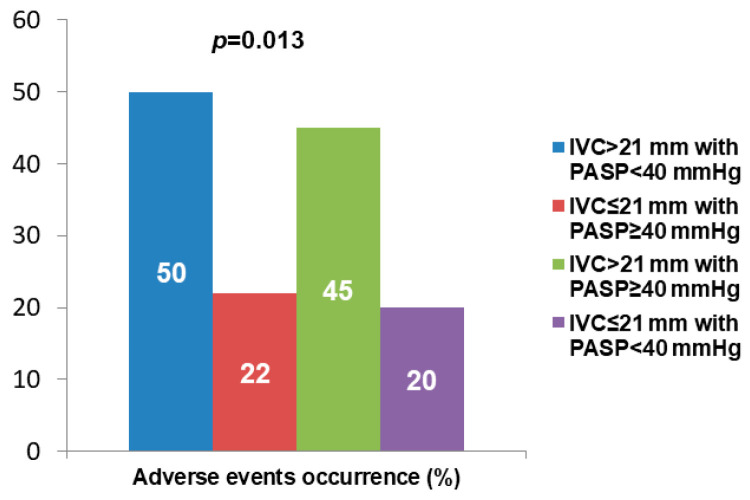
Adverse events rate according to ICV and PASP values.

**Figure 3 diagnostics-13-00692-f003:**
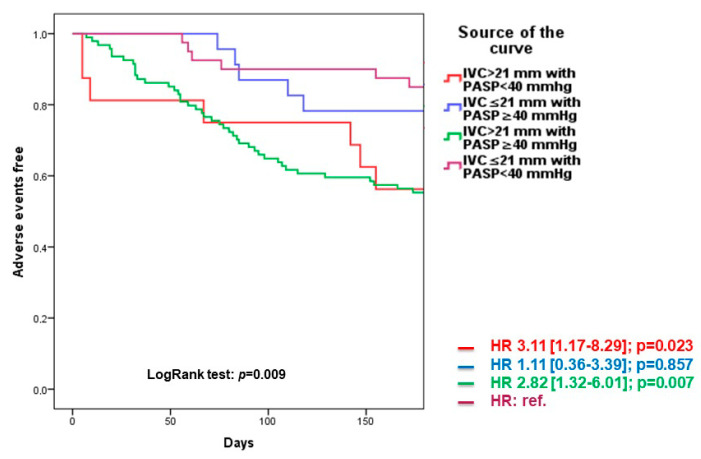
Kaplan–Meier survival curves for outcome prediction in ICV and PASP subgroups.

**Table 1 diagnostics-13-00692-t001:** Baseline characteristics of our population stratified according to adverse event occurrence.

Characteristic	All Patients (*n* = 173)	Adverse Events (*n* = 63)	No Adverse Events (*n* = 110)	*p*-Value
*Age* (years)	81 [77–86]	82 [77–87]	80 [77–85]	0.230
*Male gender* (%)	52	55	31	0.587
*Risk factors* (%):				
CAD	20	19	21	0.769
Diabetes	44	46	43	0.760
Dyslipidemia	40	35	44	0.261
Hypertension	73	67	76	0.168
Smoking	27	30	24	0.421
*Atrial Fibrillation* (%)	61	63	59	0.568
*LVEF* (%)	55 [50–57]	55 [50–55]	55 [50–57]	0.430
*LV Diameters:*				
EDD (mm)	49 [44–53]	50 [46–54]	49 [44–53]	0.423
ESD (mm)	33 [29–37]	34 [30–38]	32 [28–36]	0.243
*Volumes:*				
EDVi (ml/min^2^)	59 [55–68]	59 [55–70]	59 [55–68]	0.753
ESVi (ml/min^2^)	28 [25–32]	28 [25–32]	28 [25–32]	0.609
*Left Atrium Area* (cm^2^)	26 [22–28]	27 [23–28]	25 [22–29]	0.278
*Left Atrium diameter* (mm)	46 [43–51]	47 [44–52]	45 [42–51]	0.269
*PASP* (mmHg)	45 [35–55]	50 [35–55]	40 [35–48]	0.005
*TAPSE* (mm)	20 [17–22]	18 [15–20]	20 [18–22]	<0.001
*RV EDD diameter* (mm)	39 [36–44]	43 [38–47]	37 [36–42]	<0.001
*Inferior Cava vein* (mm)	22 [20–24]	24 [22–25]	22 [20–23]	<0.001
*Septal thickness* (mm)	13 [12–14]	12 [12–14]	13 [12–14]	0.290
*Posterior wall* (mm)	12 [11–13]	12 [10–13]	12 [11–13]	0.274
*E/e’*	14 [12–16]	16 [14–18]	14 [12–15]	<0.001
*Admission BNP* (pg/mL)	668 [373–987]	681 [411–1330]	668 [356–932]	0.455
*Congestion score*	1.96 ± 0.75	2.38 ± 0.83	1.72 ± 0.57	<0.001

**Table 2 diagnostics-13-00692-t002:** Univariate and multivariable analysis for 180 days outcome prediction.

CV DEATH OR HF RE-HOSPITALIZATION (180 Days)				
	Univariate		Multivariable ^1^	
Parameters	HR (95% CI of HR)	*p*-Value	HR ^1^ (95% CI of HR)	*p*-Value
PASP ≥ 40 mmHg	1.60 [0.91–2.82]	0.106	0.94 [0.47–1.89]	0.874
Inferior Cave vein > 21 mm	2.75 [1.49–5.07]	0.001	3.22 [1.58–6.55]	0.001
Congestion score≥2	4.03 [2.15–7.57]	<0.001	2.35 [1.12–4.93]	0.023

^1^ Adjusted for age, gender, cardiovascular risk factors (hypertension, dyslipidemia, diabetes, coronary artery disease, smoke), atrial fibrillation, left ventricular ejection fraction, E/e’ ≥ 15 and admission BNP values.

## Data Availability

Data will be available after specific request.

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
