# Peer review of "Pulmonary Artery Systolic Pressure and Cava Vein Status in Acute Heart Failure with Preserved Ejection Fraction: Clinical and Prognostic Implications"

_diagnostics, 2023, doi:10.3390/diagnostics13040692_

Round 1

Reviewer 1 Report

More data about pts characteristics are suggested.There is no enough data about the number of patients with congestion score >2. It is not clear how many of them had PASP >40mm Hg and IVC > 21mm. In the circumstances with small numbers in the each group, there is no arguments for prediction assessment of adverse cardiac events. 

Author Response

REVIEWER 1

More data about pts characteristics are suggested.There is no enough data about the number of patients with congestion score >2. It is not clear how many of them had PASP >40mm Hg and IVC > 21mm. In the circumstances with small numbers in the each group, there is no arguments for prediction assessment of adverse cardiac events. 

Response: Thank you for the precious comment. We added the following sentence in the results section: “The prevalence of patients with congestion score ≥ 2 or <2 in patients with PASP ≥ 40 mmhg (67% vs 33%; p=0.001)  and in patients with ICV > 21 mm (68% vs 32%; p=0.001) was showed in Figure 1.”. We also added a new Figure 1 showing these findings. Moreover, we inserted mean congestion score in table 1 evaluating patients with or without adverse events occurrence.

Reviewer 2 Report

Gaetano et al in this manuscript describes the utilization of IVC distention and PASP for heart failure patients. This is well conducted study however has the presentation can be improved.  

I have the following suggestions:

1.       The abbreviation of PAPS for “pulmonary artery pressure” does not make sense. The correct terminology is PASP which is pulmonary artery systolic pressure.

2.       Similarly, IVC is inferior vena cava and not inferior cava vein.

3.       What Is VCI??

4.       Heart failure and preserved ejection fraction ( HFpEF)  - correct terminology is Heart failure with  preserved ejection fraction ( HFpEF)

5.       Please make all terms/abbreviations uniform across the whole manuscript.

6.       English language needs editing significantly, would recommend having a native English speaker review the manuscript before resubmission

a.       For example - well recognized items of increased risk in patients ; items is not correct usage

7.       Would recommend highlighting this recent study in JACC - Prognostic Impact of Echocardiographic Congestion Grade in HFpEF With and Without Atrial Fibrillation which also studied the utilization of IVC diameter for congestion.

Author Response

Gaetano et al in this manuscript describes the utilization of IVC distention and PASP for heart failure patients. This is well conducted study however has the presentation can be improved.  

I have the following suggestions:

  1. The abbreviation of PAPS for “pulmonary artery pressure” does not make sense. The correct terminology is PASP which is pulmonary artery systolic pressure.

Response: Thank you for the comment. We corrected all the abbreviations in “PASP”.

  1. Similarly, IVC is inferior vena cava and not inferior cava vein.

Response: Thank you for the comment. We corrected all the abbreviations in “ICV”.

  1. What Is VCI??

Response: Thank you for the comment. We corrected all the abbreviations in “ICV”.

  1. Heart failure and preserved ejection fraction ( HFpEF)  - correct terminology is Heart failure with  preserved ejection fraction ( HFpEF)

Response: Thank you for the comment. We corrected it in  “Heart failure with  preserved ejection fraction ( HFpEF)”.

  1. Please make all terms/abbreviations uniform across the whole manuscript.

Response: Thank you for the comment. We corrected all the abbreviations in the manuscript.

  1. English language needs editing significantly, would recommend having a native English speaker review the manuscript before resubmission
  2. For example - well recognized items of increased risk in patients ; items is not correct usage

Response: Thank you for the comment. We changed the sentence as follows: “Right ventricular (RV) dysfunction and pulmonary hypertension are well recognized items offactors related to the  increased risk of adverse events occurrence in patients affected by acute heart failure (AHF)”

  1. Would recommend highlighting this recent study in JACC - Prognostic Impact of Echocardiographic Congestion Grade in HFpEF With and Without Atrial Fibrillation which also studied the utilization of IVC diameter for congestion.

Response: Thank you for the comment. We inserted the suggested reference in the discussion section under the number 39.

Reviewer 3 Report

The manuscript deals with an interesting issue and adds new data on the diagnostic handling of acute patients with heart failure and preserved ejection fraction (HFpEF). As the authors state in the limitations section, the amount of retrospectively examined patients is low. However, the conclusion of their data analysis is valuable for the diagnostic handling of this specific patient group.

The authors must review the whole manuscript and correct numerous faults in abbreviations (IVC, VC etc.). When talking about the "inferior cava vein" the abbreviation IVC should be used continously.

The letter size in figure 2 "source of the curves" should be increased.

Author Response

The manuscript deals with an interesting issue and adds new data on the diagnostic handling of acute patients with heart failure and preserved ejection fraction (HFpEF). As the authors state in the limitations section, the amount of retrospectively examined patients is low. However, the conclusion of their data analysis is valuable for the diagnostic handling of this specific patient group.

The authors must review the whole manuscript and correct numerous faults in abbreviations (IVC, VC etc.). When talking about the "inferior cava vein" the abbreviation IVC should be used continously.

Response: Thank you for the comment. We corrected all the abbreviations in “inferior cave vein (ICV)” and in “pulmonary artery systolic pressure (PASP)”.

The letter size in figure 2 "source of the curves" should be increased.

Response: Thank you for the comment. We corrected the figure 3 (old figure 2) as suggested.

Reviewer 4 Report

The study carried out by your team is interesting and has elements of originality as well as scientific rigor, so I congratulate you for the work done.

On another note, I notice a very high concentration of figures resulting from echocardiographic measurements, otherwise, well presented in tables and graphs, but still too many and difficult to follow. I must also add the fact that the study tries to expose in the echocardiographic parameters the early pathophysiological mechanism of heart failure and to identify through the same echocardiographic parameters the evolutionary possibilities of an acute heart failure. As you also show in the study, one of the parameters failed in the attempt to be a predictor of the evolution of acute heart failure with preserved ejection fraction.

I would be of the opinion that this study would have a greater scientific value if it were carried out in parallel with an imaging evaluation of the heart through magnetic resonance and later, the echocardiographic results would be compared with the results obtained through magnetic resonance.

Under these conditions, a single parameter is insufficient to be taken into account as a predictor of evolution.

The conclusions from the study are insufficient to bring an increased argumentative value compared to the clinical-biological data.

Summarizing, I could say that the study is too complex compared to the conclusions obtained, and for current medical practice it is too laborious to actually demonstrate what is already known regarding the pathophysiological changes and evolutionary possibilities of an acute heart failure.

Author Response

The study carried out by your team is interesting and has elements of originality as well as scientific rigor, so I congratulate you for the work done.

On another note, I notice a very high concentration of figures resulting from echocardiographic measurements, otherwise, well presented in tables and graphs, but still too many and difficult to follow. I must also add the fact that the study tries to expose in the echocardiographic parameters the early pathophysiological mechanism of heart failure and to identify through the same echocardiographic parameters the evolutionary possibilities of an acute heart failure. As you also show in the study, one of the parameters failed in the attempt to be a predictor of the evolution of acute heart failure with preserved ejection fraction.

I would be of the opinion that this study would have a greater scientific value if it were carried out in parallel with an imaging evaluation of the heart through magnetic resonance and later, the echocardiographic results would be compared with the results obtained through magnetic resonance.

Response: Thank you for the comment. We agree with you on the importance of cardiac magnetic resonance, however it would be very difficult to study acute HFpEF with this imaging technique due to high heart rate and high respiratory rate during AHF phases . However we inserted this concept in the limitation section.

Under these conditions, a single parameter is insufficient to be taken into account as a predictor of evolution.

The conclusions from the study are insufficient to bring an increased argumentative value compared to the clinical-biological data.

Summarizing, I could say that the study is too complex compared to the conclusions obtained, and for current medical practice it is too laborious to actually demonstrate what is already known regarding the pathophysiological changes and evolutionary possibilities of an acute heart failure.

For sure a more complete echo and lung ultrasound assessment may confer much more information, we added this concept in Limitations. however in clinical practice patients AHF is often treated before to know congestion and perfusion status with diuretics and vasodilators or inotropes in relation on blood pressure values. Indeed the an early echocardiographic assessment could be done in each admitted patients with acute HF diagnosis in order to better understand the type of congestion and primitive cardiac alterations. The measurement of pulmonary pressure and cava vein, provide a non invasive estimation useful for clinicians. Our final message is in accordance with this position.

Round 2

Reviewer 4 Report

I have analyzed the improved version of the article and found that you managed to meet the reviewers' requirements to some extent. Although, currently, echocardiography dominates the imaging examination of the heart, it would be desirable to mention in your article some brief references to the importance of MRI in the future of cardiac imaging and functional examinations, as well as some references to the determination of pressures in the right ventricle, i.e. the method of determining them and their influence on the presumptives in the pulmonary artery. Also, I think that the factors that influence the pressure in the inferior vena cava are worth mentioning. Finally, re-reading the title, I found that no reference is made to the situations in which acute cardiac insufficiency occurs in patients with heart failure with preserved ejection fraction. In these conditions of acute heart failure, can it still be a matter of a preserved ejection fraction!?

A clearer statement regarding the relationship between PASP and ICV dilatation would be desirable. Is the relationship predictive of acute heart failure in patients with HFpEF or does it appear after the exacerbation has occurred?

From the echocardiographic point of view, things seem simple, but from the point of view of pathogenic and physiopathological mechanisms, the situation is much more complicated.

Author Response

I have analyzed the improved version of the article and found that you managed to meet the reviewers' requirements to some extent. Although, currently, echocardiography dominates the imaging examination of the heart, it would be desirable to mention in your article some brief references to the importance of MRI in the future of cardiac imaging and functional examinations, as well as some references to the determination of pressures in the right ventricle, i.e. the method of determining them and their influence on the presumptives in the pulmonary artery.

Many thanks for your further advices and suggestion. We reported the restricted role of cardiac MRI in acute HF because of long time execution and supine position that acute patients do not tolerate for prolonged period. Indeed these patients often take O2 supply and infusional therapy. Under this condition it becomes hard to execute MRI. For your agreement we inserted in the reference list a paper ( ref 42) and we reported these concepts in limitations paragraphs.

Also, I think that the factors that influence the pressure in the inferior vena cava are worth mentioning. Finally, re-reading the title, I found that no reference is made to the situations in which acute cardiac insufficiency occurs in patients with heart failure with preserved ejection fraction. In these conditions of acute heart failure, can it still be a matter of a preserved ejection fraction!?

 This is an interesting point. Literature did not report significant difference among acute HFrEF and HFpEF in terms of signs and symptomps of congestion, however the effects of different hemodynamic profile and pathophysiology may influence the echo findings- The novelty of our paper consists in detailed study of this setting poorly investigated up to now. Accordingly We added the following sentences in discussion “Nevertheless, a study comparing echocardiographic findings of congestion in acute HFpEF vs HFrEF showed a similar profile in terms of RV dysfunction and venous congestion. (32) Conversely a large cohort trial measuring pulmonary pressure in all HF categories, revealed that elevated PASP is significantly related to readmission in HFpEF patients ( 33).

A clearer statement regarding the relationship between PASP and ICV dilatation would be desirable. Is the relationship predictive of acute heart failure in patients with HFpEF or does it appear after the exacerbation has occurred?

This question is hardly answerable, since we just found a significant relation during acute phase. It could be probable that in stable condition with optimal decongestion the two parameters are not related together.We added this point in Limitation

From the echocardiographic point of view, things seem simple, but from the point of view of pathogenic and physiopathological mechanisms, the situation is much more complicated.

To be honest, I don’t know what the reviewer means with this statement, our paper is just a picture of echocardiographic findings in acute HFpEF setting revealing both pulmonary and venous pressure status by non invasive tool

Round 3

Reviewer 4 Report

I read your answers and I think they were prompt and reasoned. The article in its current form can be published.